# Network medicine for disease module identification and drug repurposing with the NeDRex platform

Sepideh Sadegh 1,2,12✉, James Skelton3,12, Elisa Anastasi3, Judith Bernett 1, David B. Blumenthal 4, Gihanna Galindez5,6, Marisol Salgado-Albarrán1,7, Olga Lazareva1, Keith Flanagan3, Simon Cockell8, Cristian Nogales9, Ana I. Casas9,10, Harald H. H. W. Schmidt 9, Jan Baumbach 2,11,13, Anil Wipat3,13 & Tim Kacprowski 5,6,13

Traditional drug discovery faces a severe efficacy crisis. Repurposing of registered drugs provides an alternative with lower costs and faster drug development timelines. However, the data necessary for the identification of disease modules, i.e. pathways and sub-networks describing the mechanisms of complex diseases which contain potential drug targets, are scattered across independent databases. Moreover, existing studies are limited to predictions for specific diseases or non-translational algorithmic approaches. There is an unmet need for adaptable tools allowing biomedical researchers to employ network-based drug repurposing approaches for their individual use cases. We close this gap with NeDRex, an integrative and interactive platform for network-based drug repurposing and disease module discovery. NeDRex integrates ten different data sources covering genes, drugs, drug targets, disease annotations, and their relationships. NeDRex allows for constructing heterogeneous biological networks, mining them for disease modules, prioritizing drugs targeting disease mechanisms, and statistical validation. We demonstrate the utility of NeDRex in five specific use-cases.

1 Chair of Experimental Bioinformatics, TUM School of Life Sciences, Technical University of Munich, Munich, Germany. 2 Chair of Computational Systems Biology, University of Hamburg, Hamburg, Germany. 3 School of Computing, Newcastle University, Newcastle upon Tyne, UK. 4 Department Artificial Intelligence in Biomedical Engineering, Friedrich-Alexander University Erlangen-Nürnberg, Erlangen, Germany. 5 Division Data Science in Biomedicine, Peter L. Reichertz Institute for Medical Informatics of Technische Universität Braunschweig and Hannover Medical School, Braunschweig, Germany. 6 Braunschweig Integrated Centre of Systems Biology (BRICS), TU Braunschweig, Braunschweig, Germany. 7 Natural Sciences Department, Universidad Autónoma Metropolitana-Cuajimalpa, Mexico City, Mexico. 8 School of Biomedical, Nutrition and Sports Sciences, Faculty of Medical Sciences, Newcastle University, Newcastle upon Tyne, UK. 9 Department of Pharmacology and Personalised Medicine, School for Mental Health and Neuroscience (MHeNs), Maastricht University, Maastricht, the Netherlands. 10 Department of Neurology, University Hospital Essen, Essen, Germany. 11 Computational Biomedicine Lab, Department of Mathematics and Computer Science, University of Southern Denmark, Odense, Denmark. 12 These authors contributed equally: Sepideh Sadegh, James Skelton. 13 These authors jointly supervised this work: Jan Baumbach, Anil Wipat, Tim Kacprowski. ✉email: sadegh@wzw.tum.de

Between 1950 and 2010, the productivity of drug development halved approximately every 9 years[1]. Although this trend has changed over the past ten years[2], the cost of bringing a new molecular entity to market is still estimated to be between two and three billion USD[3]. Contributing factors to these high costs include a plethora of already effective treatments, irreproducibility of pre-clinical research and an increase of caution amongst drug regulatory agencies[1]. Consequently, there is interest in alternative approaches to finding therapeutics.

Drug repurposing, also known as drug repositioning, is the process of identifying alternative uses for existing drugs. In comparison to traditional drug development, drug repurposing offers significant advantages such as low cost, reduced risk, and faster drug development timelines. While early examples of successfully repurposed drugs have been identified through serendipitous discoveries, advances in omics technologies and the availability of massive amounts of omics data have provided opportunities for systematic in silico inference of new drug-disease relationships.

Various in silico drug repurposing strategies have been proposed, including signature-, knowledge-, network-, and machine learning-based approaches[4]. Network-based approaches are particularly attractive, because networks offer a natural representation of complex biological associations and provide a framework for incorporating multiple data types. In such networks, nodes can represent drugs, proteins, or diseases, and edges indicate drug-drug similarities, drug-target interactions, gene-disease associations, and gene-gene interactions (e.g., protein-protein interaction (PPI) networks, gene regulatory networks, signaling networks, and metabolic networks)[5].

Moreover, previous studies have indicated that disease-associated genes are not randomly scattered throughout biological networks. Instead, they tend to be located in so-called disease modules, i.e., small subnetworks representing interconnected mechanisms that can be linked to the phenotype[6–8]. One of the guiding paradigms of network-based drug repurposing is that diseases can be viewed as perturbations of these modules[8]. Consequently, potentially repurposable drugs can be identified in silico by carrying out the following three steps:

1. Construct a heterogeneous biological network by integrating data from multiple biomedical databases which are relevant for the given task.
2. Mine the constructed biological network to derive disease modules associated with the disease of interest.
3. Extract prioritized list of drugs whose known targets are contained in or situated in close vicinity of the extracted disease modules.

Network-based drug repurposing is a highly active field of research, which has been boosted even further with the advent of the COVID-19 pandemic. However, studies have so far been limited to presenting either non-translational algorithmic results or specific predictions limited to certain diseases. There is still an urgent need for integrated tools which allow experts from pharmacology or biomedical research fields to easily carry out all three steps of network-based drug repurposing and adapt them to the needs of their individual use cases. To the best of our knowledge, the only available tools that begin to address this need are Hetionet[5] and CoVex[9]. However, Hetionet is static and only allows the user to browse for pre-computed results related to a fixed set of 136 diseases (algorithms are provided only as separate Python packages and are not integrated into the platform). CoVex does allow the user to interact with the system, but it is limited to COVID-19 drug repurposing.

We present the NeDRex (Network-based Drug Repurposing and exploration) platform—a generically applicable integrated platform for network-based disease module discovery and drug repurposing. Figure 1 illustrates the overview of the platform. NeDRex is built of three main components: a knowledgebase (NeDRexDB, available at http://neo4j.nedrex.net/ and https://api.nedrex.net/), a Cytoscape app (NeDRexApp, available at https://apps.cytoscape.org/apps/nedrex), and an API (NeDRex-API, available at https://api.nedrex.net/).

NeDRexDB integrates data from various biomedical databases such as OMIM[10], DisGeNET[11], UniProt[12], NCBI gene info[13], IID[14], MONDO[15], DrugBank[16], Reactome[17], and DrugCentral[18]. Integration of multiple databases enables us to construct heterogeneous networks representing distinct types of biomedical entities (e.g., diseases, genes, drugs) and the associations between them. These networks can be accessed and explored via NeDRexApp, NeDRexAPI, and the Neo4j endpoint to NeDRexDB. For more details on the different types of integrated data in NeDRexDB, see Supplementary Table 1, 2 and Supplementary Fig. 1.

NeDRexApp is a Cytoscape app[19] that provides implementations of state-of-the-art network algorithms, such as Multi-Steiner Trees (MuST)[9], TrustRank[20], Biclustering Constrained by Networks (BiCoN)[21], and Disease Module Detection (DIAMOnD)[8]. These functionalities are made available to the user via the RESTful API and the easy-to-use NeDRexApp. All algorithms require a list of user-selected genes (referred to as seeds) as the starting point, except for BiCoN, which uses gene expression data. Seeds can be all or a subset of the genes associated with the disease, so-called disease genes, or genes contained in disease modules. Moreover, expert knowledge can be employed for seed selection, and the results can be statistically validated by calculating the empirical $P$ values (Fig. 1). NeDRex, hence, allows researchers from pharmacology and biomedicine to leverage their expert knowledge for discovering drug repurposing candidates via state-of-the-art network medicine methods. In particular, our platform can also be used to identify disease modules and possibly repurposable drugs for any newly discovered disease such as COVID-19.

The remainder of the paper is organized as follows: In the Results section, we first provide an overview of the NeDRex platform. Subsequently, we present several use cases which exemplify how to use NeDRex for disease module identification and drug repurposing. In the Discussion section, we discuss prospects and limitations of using NeDRex for drug repurposing. In the Methods section, we describe the datasets and the integration scheme used in NeDRexDB. We also introduce the logic behind the network medicine algorithms implemented in NeDRex, and briefly describe the general architecture of the platform.

## Results

**The NeDRex platform.** The main result is the NeDRex platform itself, which provides a broad spectrum of systems medicine methods together with integrative networks of different biological entities. The platform is modular and new algorithms and databases can be easily incorporated. In addition, the NeDRexDB knowledgebase, which is accessible via the RESTful API and Neo4j endpoint, serves as a useful resource for scientists to explore the relationships between different biological entities, such as drugs, diseases, genes, proteins, and pathways. Moreover, by using NeDRexApp, users can build custom networks from the NeDRexDB knowledgebase according to their needs and further explore them via the various network medicine functionalities (the complete list of functionalities is available in the tutorial document of the app: https://nedrex.net/tutorial). Finally, users can also download the data from NeDRexDB and employ it for

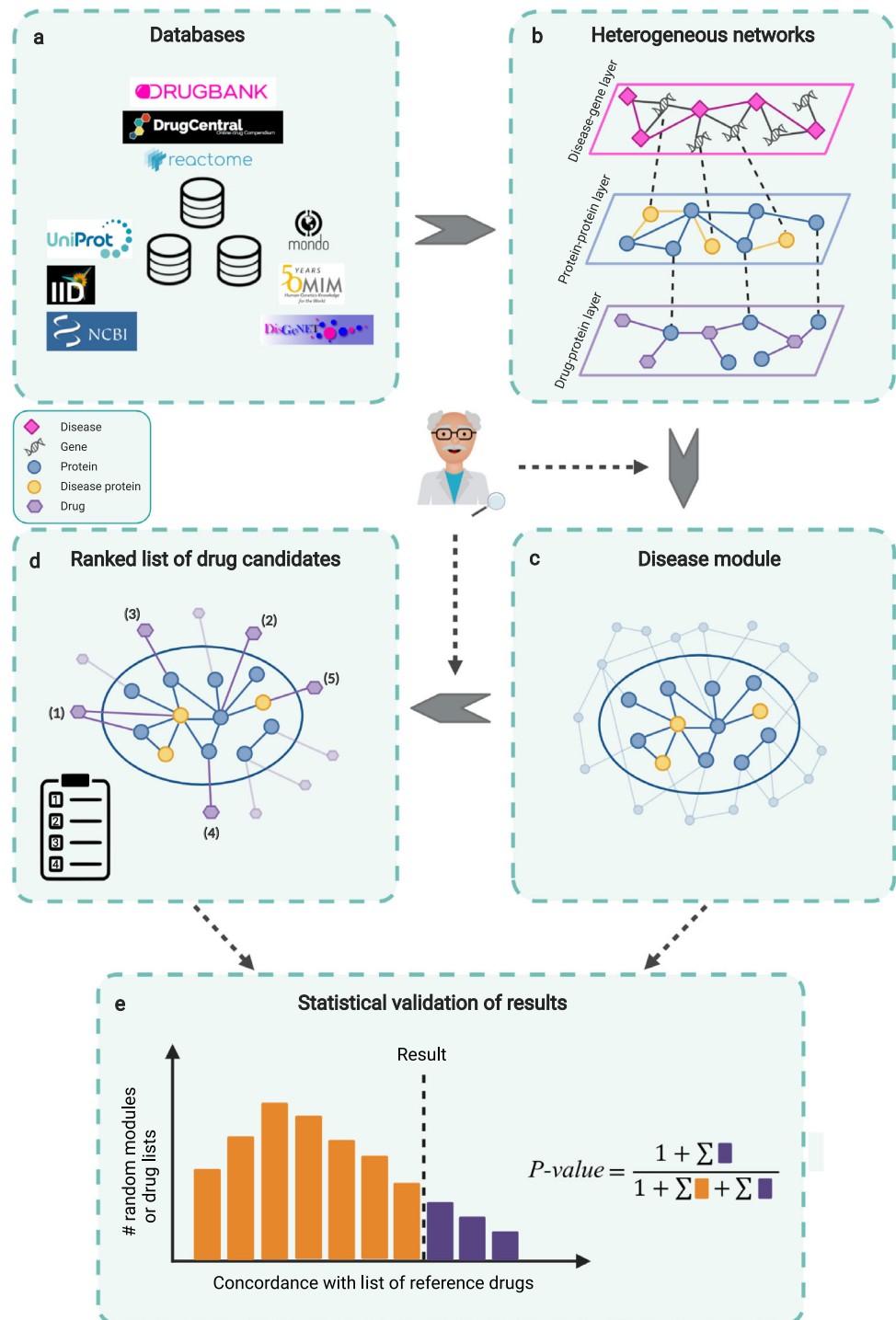

**Fig. 1 Overview of the NeDRex platform. a** Integration of various biomedical databases. **b** Construction of heterogeneous networks. **c** Disease module identification using network-based algorithms (MuST, DIAMOnD, BiCoN). **d** Ranking of drugs using network-based algorithms (TrustRank, closeness centrality). Benefiting from the expert-in-the-loop paradigm, expert knowledge can be engaged at two points: (1) before the disease module identification step through selecting seeds; (2) before the drug ranking step through selecting seeds for ranking algorithms. **e** Statistical validation of the obtained disease modules and ranked lists of drugs via empirical *P* values. X-axis: Concordance of contained drugs (for drug list validation) or targeting drugs (for disease module validation) with list of reference (e.g., indicated) drugs. Created with BioRender.com.

their own drug repurposing methods. Table 1 provides an overview of the main functionalities provided by NeDRex.

The typical steps users should take in NeDRexApp to derive disease modules and pinpoint drug candidates starting with the disease(s) under study are illustrated in Fig. 2. For more information about seed selection, see Supplementary Information.

For more details on the algorithms, the selected seeds, the parameters applied for each use case and their statistical validation, see Methods and Supplementary Information (Result section), respectively. In the following, we demonstrate the applicability of NeDRex in five different use cases employing a variety of available functionalities. Detailed tutorials to reproduce

**Table 1 Overview of the main functionalities of the NeDRex platform.**

| Functionality | Description |
|---|---|
| Integrating data from multiple biomedical databases | NeDRexDB is an integrated knowledgebase which is accessible via NeDRexAPI as well as a Neo4j endpoint. |
| Constructing heterogeneous biological networks from NeDRexDB | Based on users' needs, different heterogeneous networks can be constructed using NeDRexAPI or NeDRexApp. |
| Disease module mining | Various disease module identification algorithms can be run on NeDRexDB using NeDRexApp or NeDRexAPI, based on users' inputs. |
| Drug prioritization | Various drug prioritization algorithms can be run on NeDRexDB using NeDRexApp or NeDRexAPI, based on users' inputs and the results of disease module mining. |
| Statistical validation of results | The results of disease module identification and drug prioritization analyses can be validated with different statistical methods. |
| Visualization of results | Using NeDRexApp, all the obtained results are shown in network format, which can be explored further. |

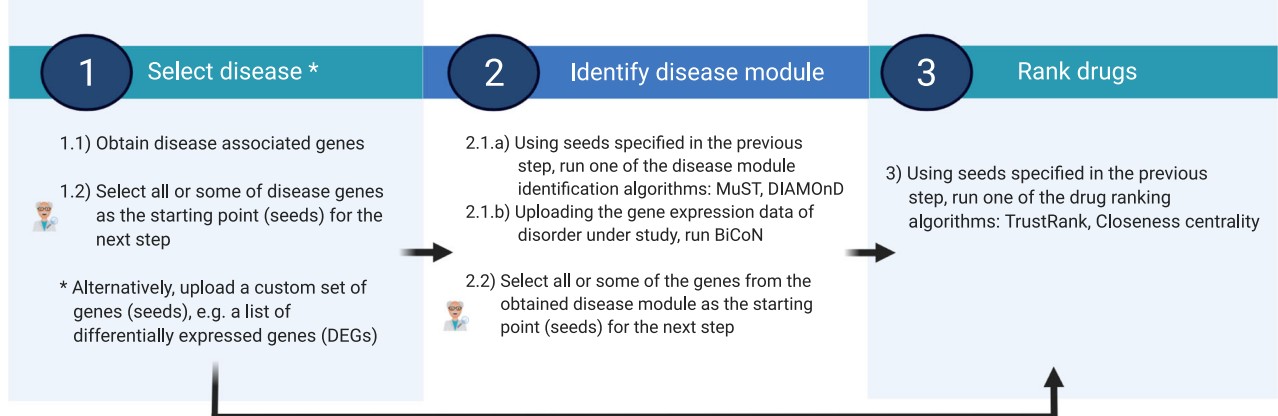

**Fig. 2 Typical steps in NeDRexApp to identify disease modules and drug candidates.** (Step 1) The workflow can start either with selecting the disease(s) under study and subsequently obtaining genes associated with them or uploading a custom set of genes, e.g., DEGs. (Step 2) Disease modules are derived using seeds selected in the previous step as input and employing the MuST or DIAMOnD algorithm. Alternatively, the BiCoN algorithm can be employed to return disease modules. In this case, step 1 is skipped and gene expression data should be used as input for this step. (Step 3) Drugs targeting directly or the vicinity of the seeds selected in the previous step are ranked. Step 3 can also be performed directly after step 1. Expert knowledge can be involved at seed selection points 1.2 and 2.2. Created with BioRender.com.

the use cases with NeDRexApp are available at https://nedrex.net/tutorial. Note that the results obtained for the use cases constitute hypotheses which have not been further experimentally validated. The main purpose of the use cases is to exemplify how to use the rich functionality available in NeDRex.

**Use case 1: identification of disease pathways for ovarian cancer, using MuST.** To exemplify the power of NeDRex to extract biologically meaningful pathways from starting seeds, we used the ovarian cancer (OC) associated genes from NeDRexDB (*AKT1, ALPK2, CDH1, CTNNB1, EPHB1, OPCML, PIK3CA, PRKN*) and constructed disease module using the MuST algorithm (Fig. 3.a). The obtained disease module contains newly identified connector genes (*ATXN1, HTT, HSP90AA1, PDGFRB, NCK1, OLA1* and *DKK3*) which, together with the seed nodes, participate in relevant OC pathways that could not be retrieved using the seed genes alone. In particular, genes involved in ovary-specific, hormone-related and cancer pathways are found (Fig. 3b). For instance, using the g:Profiler enrichment tool and the KEGG pathway database[22,23], we find that the OC module is enriched in the progesterone-mediated oocyte maturation and the Estrogen signaling pathway, which are both involved in oocyte maturation[24]. Furthermore, we find that the ErbB signaling pathway, which is involved in cancer cell growth, proliferation, motility, and survival[25] is associated with the disease module. We also identified further cancer-related pathways, namely, choline

metabolism in cancer, central carbon metabolism in cancer, and EGFR tyrosine kinase inhibitor resistance[26–28]. Finally, the examination of the connector genes identified by MuST reveals the *PDGFRB* gene, which has been reported to be deregulated in 40–80% of ovarian tumors[29,30] and has been proposed as a therapeutic target in OC[31].

Together, these results show that, using MuST, NeDRex was capable of identifying a disease module containing genes associated with meaningful biological pathways. Notably, although the number of seeds and the size of the disease module is small, we found ovary-specific and cancer-associated pathways, as well as genes involved in OC.

**Use case 2: identification of therapeutic drugs for inflammatory bowel disease, using MuST and drug ranking algorithms.** To demonstrate the utility of the NeDRex platform to recover known and potential therapeutic drugs, we selected inflammatory bowel disease (IBD). Using the Get Disease Genes function, all the known genes associated with IBD are obtained from NeDRexDB. Running the MuST algorithm starting with this set of genes as seeds outputs a disease module containing 87 genes, which are targeted by a total of 235 drugs (empirical precision-based *P* value: 0.036). Considering the high number of drugs targeting this module, the user can prioritize the most promising candidates by using one of the drug ranking functionalities. After running the closeness centrality algorithm, three small molecules

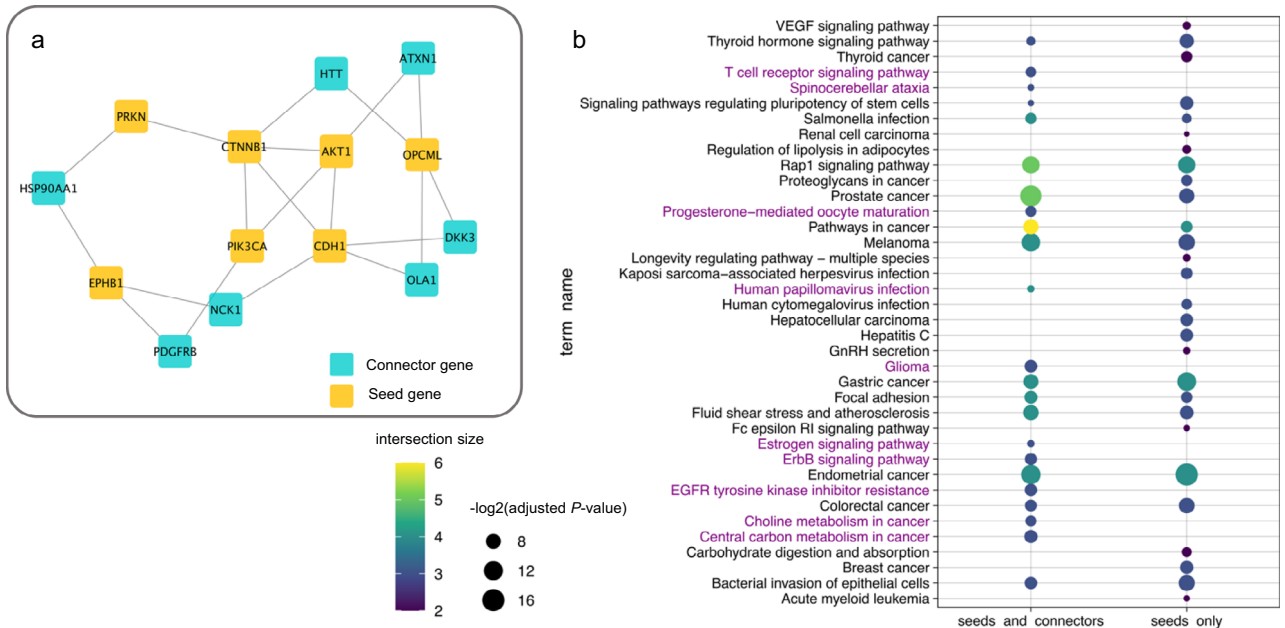

**Fig. 3 Ovarian cancer disease pathway identification by MuST. a** The OC disease module derived by MuST using NeDRexDB OC-associated genes (seeds). **b** Comparison of KEGG enriched pathways obtained with seed and connector genes vs. obtained using seed genes alone. Pathways which could only be retrieved after adding connector genes are marked in purple.

among the top-ranked drugs, namely, Fostamatinib (1), Ruxolitinib (5), and Imatinib (12) are identified, whose relevance to IBD is supported by literature evidence[32–35]. The IBD disease module together with the 25 top-ranked drugs targeting the module is shown in Supplementary Fig. 2. Imatinib therapy has been reported to induce remission in IBD patients[32]. Fostamatinib was reported to alleviate IBD-induced inflammatory damage in rats[33]. The JAK inhibitor Ruxolitinib has been reported to ameliorate ulcerative colitis in a mouse model[35].

The DCG-based empirical $P$ value of the ranked list of drugs computed via closeness centrality is <0.001. The joint validation of the obtained disease module and drug list yielded a precision-based empirical $P$ value of <0.001. Overall, these results provide further motivation to explore the potential of other top-ranked drugs in the treatment of IBD derived by the two algorithms using NeDRex.

**Use case 3: drug target and drug identification for pulmonary embolism, using combination of DIAMOnD and TrustRank.** Next, we demonstrate how NeDRex can uncover a pulmonary embolism (PE) disease module using the DIAMOnD algorithm and subsequently recover drugs indicated for treatment of PE. Using data from NeDRexDB, twelve genes are found to be associated with PE. When selecting all of these genes as starting seeds, the DIAMOnD algorithm returns a subnetwork of 32 genes representing the underlying mechanistic pathways for PE (precision-based empirical $P$ value: 0.012). A total of 283 drugs target this module. By employing the TrustRank algorithm to prioritize the drugs associated with the disease module (excluding the initial seeds), we find Bemiparin, Edoxaban, Apixaban, Dabigatran etexilate, Heparin, Rivaroxaban, Streptokinase, and Urokinase among the 50 top-ranked drugs. All of these drugs are indicated to reduce the risk of stroke and systemic embolism and are known to be used to treat PE. Furthermore, five drugs registered in ClinicalTrials.org for evaluation in treatment of PE, namely, Alteplase, Enoxaparin, Fondaparinux, Tenecteplase and Tranexamic acid are found on the top of the ranked list.

The PE disease module (excluding the initial seeds) combined with its targeting top-ranked drugs is shown in Fig. 4. Apixaban, Bemiparin, Dabigatran etexilate, Edoxaban, Enoxaparin, Fondaparinux, Heparin, and Rivaroxaban target the coagulation factor X (F10), which is not among the initial set of PE-associated genes but is found in the PE module. F10 is a key enzyme in the coagulation cascade[36]. Alteplase, Dabigatran etexilate, Streptokinase, Tenecteplase, Tranexamic acid, and Urokinase target plasminogen (PLG), another member of the PE disease module that helps dissolving the fibrin of blood clots and behaves as a proteolytic factor[36]. Another gene found in the PE disease module is SERPINE1, whose product (plasminogen activator inhibitor 1) is a protease inhibitor that is targeted by Alteplase, Tenecteplase, and Urokinase from the list of predicted drugs. This protein is essential for inhibiting fibrinolysis and is in charge of the controlled degradation of blood clots[37,38].

The DCG-based empirical $P$ value of the ranked list of drugs computed using TrustRank is <0.001 (precision-based $P$ value obtained by joint validation of module and drug list: 0.018). This use case indicates, firstly, that NeDRex is capable of extracting disease-related mechanistic pathways, which can contain possible targets for candidate drugs. Secondly, drugs which in practice are prescribed for treatment of PE or are under evaluation in clinical trials are among the top-ranked drugs obtained by the drug ranking algorithms.

**Use case 4: disease module and drug identification for Huntington's disease, using BiCoN and TrustRank.** BiCoN is an unsupervised approach that simultaneously performs patient and gene clustering such that the genes that provide the best possible clustering are also connected in a PPI network. We use BiCoN on Huntington's disease (HD) gene expression data from GEO (accession number GSE3790[39,40]), which contain patients with Vonsattel grades 2–4 and healthy controls (precision-based empirical $P$ value of the obtained HD disease module: 0.180). Patient clusters reported by BiCoN show strong correlation with the known phenotype (average Jaccard index 0.76), providing

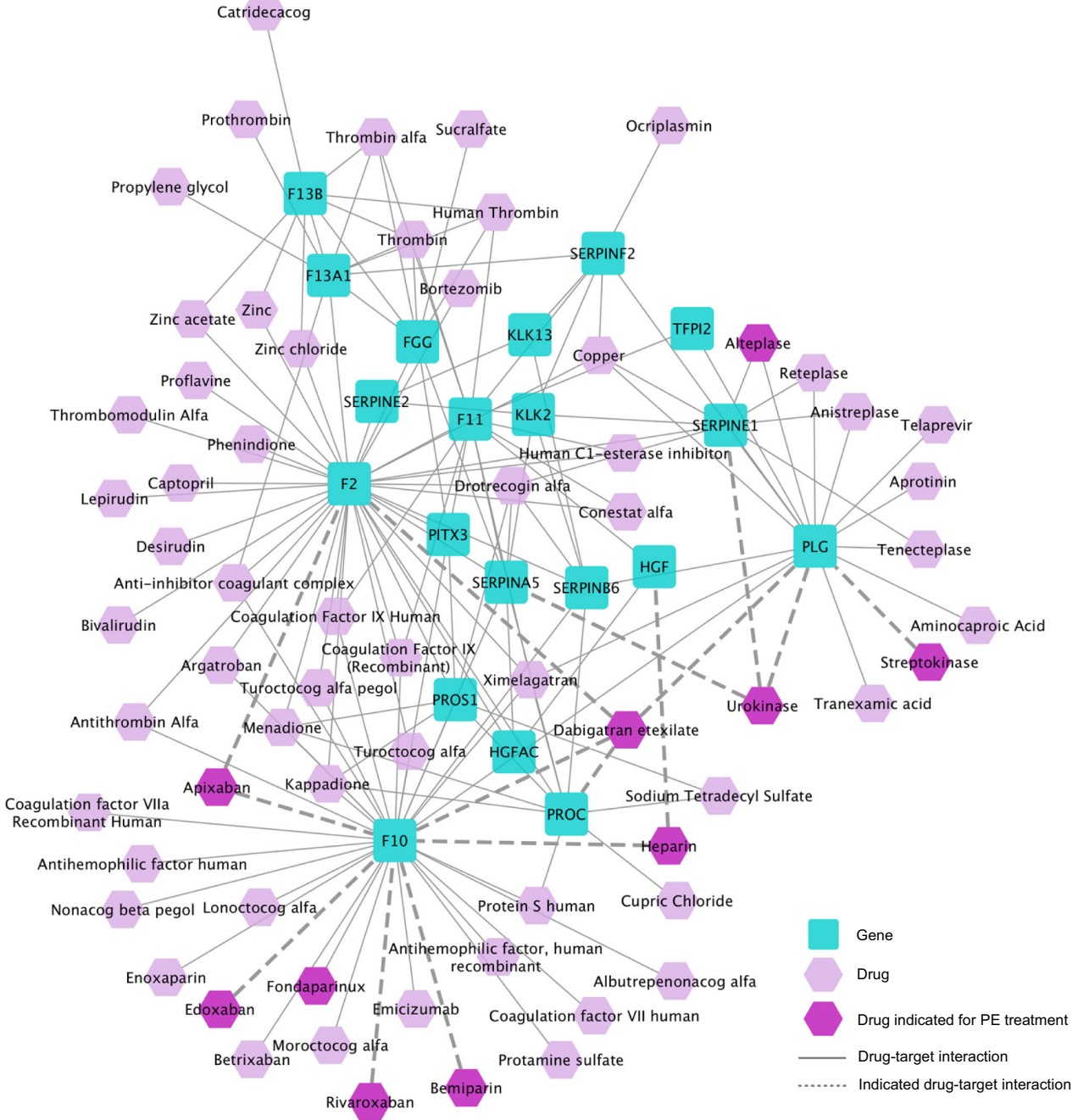

**Fig. 4 The pulmonary embolism disease module and its targeting top-ranked drugs.** The PE disease module (excluding the initial seeds) derived by DIAMOnD, combined with its targeting 50 top-ranked drugs.

strong evidence to assume that the reported subnetwork (23 genes in total) is also closely related to the disease mechanism.

We ran TrustRank on the subnetwork returned by BiCoN, and among the 50 top-ranked drugs we find three drugs that are prescribed to alleviate the symptoms of HD, namely, Thorazine (Chlorpromazine), Memantine, and Lamotrigine (Fig. 5). Thorazine is prescribed to help manage movement disorders, such as chorea in people with HD[41]. According to Beister et al.[42], memantine can slow down the progression of HD. Lamotrigine significantly improves depression, severe mood swings, and choreoathetoidic movements in HD patients[43].

Among other high scoring drugs that target the derived subnetwork and have a strong connection to HD are Donepezil[44],

Decamethonium[45], Betahistine[46] (used to treat dizziness), Fluoxetine[47] (recommended for HD patients to treat aggressiveness and agitation), Pitolisant[48] (treats narcolepsy), and other drugs that are used for treating HD patients and management of HD symptoms. DCG-based empirical $P$ value of the ranked list of drugs computed using TrustRank is 0.011 (the precision-based $P$ value obtained by joint validation of module and drug list: 0.048).

**Use case 5: hypothesis-driven drug repurposing for Alzheimer's disease.** In our last use case, we show how NeDRex can be used to extract possibly repurposable drugs which are indicated for diseases that are known to be associated with the disease of

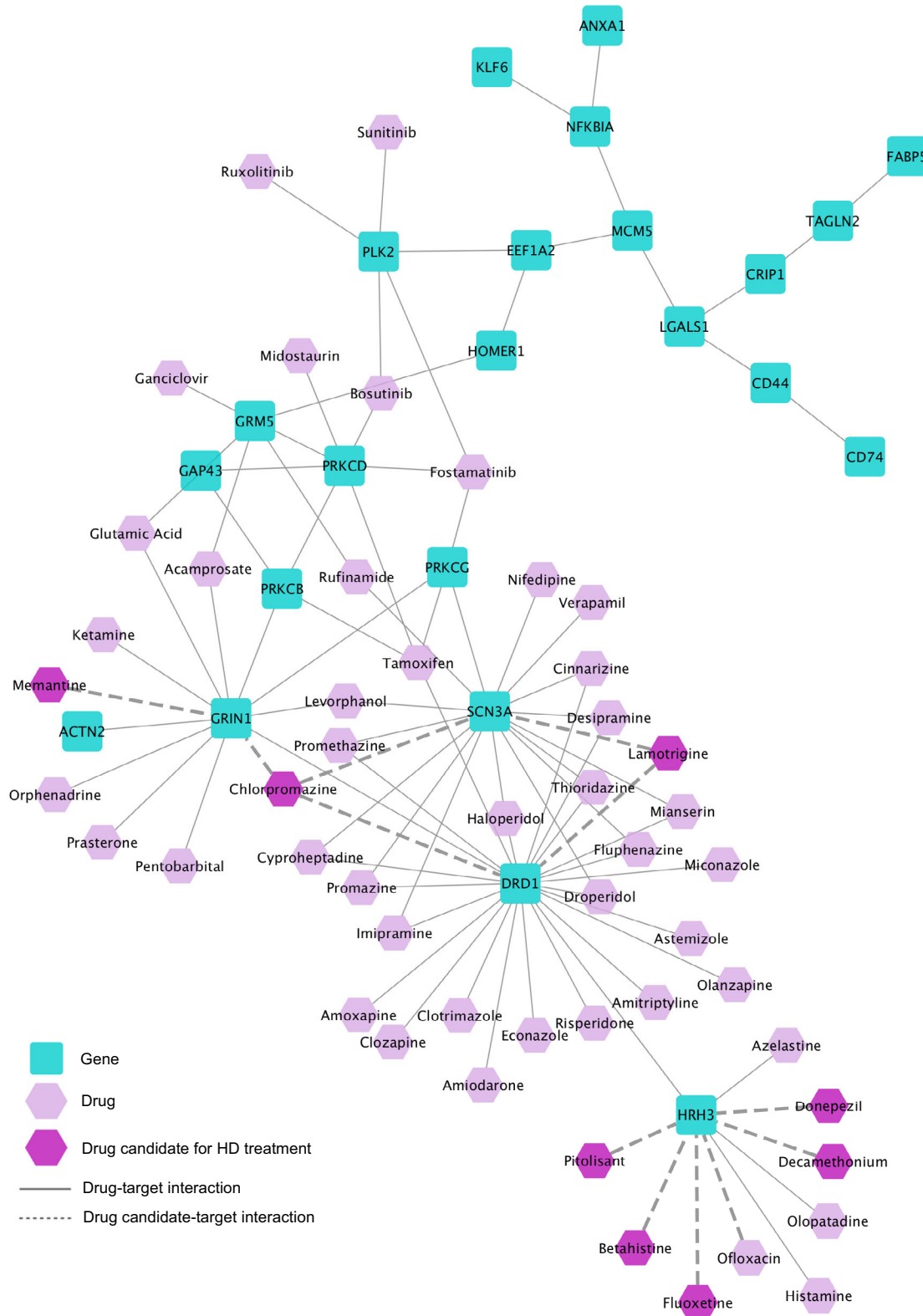

**Fig. 5 The Huntington's disease module and its targeting top-ranked drugs.** The HD disease module derived by BiCoN using gene expression data, together with its targeting 50 top-ranked drugs.

interest. More specifically, using Alzheimer's disease (AD) as an example, we show that we can retrieve potential treatments with an original indication for hypertension, diabetes mellitus (DM) and hyperlipidemia[49].

Hypertension as original indication - Here, we demonstrate how our platform can identify repurposable drugs directly from the genes associated with the new indication (AD) as a starting point. First, we obtain the genes associated with AD (40 genes). Then, we rank all the 240 drugs targeting this set of genes using the closeness centrality algorithm (DCG-based empirical $P$ value: <0.001). Interestingly, this returns Telmisartan (ranked 26th). Telmisartan is a known angiotensin II receptor blocker (ARB) originally indicated to treat high blood pressure and has been tested in clinical trials to assess its efficacy for the treatment of AD[50]. Studies show that drugs used to treat hypertension, including ARBs, decrease the risk and slow the progression of AD[51,52] by reducing the amyloid-β deposition in senile plaques, the main pathological hallmark of AD. ARBs are thought to improve amyloid-β deposition through the modulation of cerebral blood flow and superoxide production[53]. This example, hence, shows that it is possible to retrieve potentially repurposable drugs directly from the associated genes of the new indication.

Diabetes as original indication—Medications indicated for diabetes mellitus (DM) are potential treatments of AD since the glucose metabolism plays a key role in neural function[54,55]. Several drugs have been tested in vitro, in vivo and in clinical trials, where Insulin (DB00030), Insulin Detemir, Insulin Glulisine (insulin analogs) stand out[56–58]. These drugs interact with the insulin receptor (INSR) and are considered disease modifying drugs. Hence, we demonstrate that our platform is capable of retrieving this shared molecular mechanism and these drugs.

First, we obtain the DM-associated genes (88 genes), as well as the AD-associated genes (40 genes). The intersection of these sets consists of 2 genes: *INS* (whose encoded peptide, insulin, is a repurposed drug in AD) and *INSR* ($P$ value = 0.017071, hypergeometric test for overlap of two disease gene sets). NeDRexDB contains 32 drugs targeting the products of these 2 genes (overlap-based empirical $P$ value: 0.002). Notably, 27 of these drugs target INSR including repurposed drugs; such as Insulin Detemir and Insulin Glulisine. Note that, in this use case, we did not use any network algorithms to extract the drug repurposing candidates but only leveraged the data integration functionalities provided by NeDRex.

Hyperlipidemia as original indication—With this example, we show how to search for potentially repurposable drugs by retrieving drugs that indirectly target the intersection of disease modules for two diseases, namely, hyperlipidemia and AD. We use the hyperlipidemia-associated genes, since the lipid and cholesterol metabolism has been linked with progression of AD[59].

First, by using NeDRexDB, we extract the hyperlipidemia-associated genes (19 genes) and derive the disease module using DIAMOnD. Similarly, we derive the AD module starting with its associated genes (40 genes). By obtaining the intersection of the two modules, we find 7 genes in common ($P$ value of hypergeometric test = 0.023827): *A2M*, *APOE*, *APP*, *CLU*, *IGF2*, *NOS3*, and *PLAU* (precision-based empirical $P$ value of intersection: 0.079). Notably, all of them are AD-associated genes and some are well-characterized drivers of this disease; for instance, *APP* encodes the amyloid-β peptides[60], *A2M* is a marker of neural damage[61], and *APOE*, *CLU* and *NOS3* polymorphisms are risk markers of AD[62]. Importantly, *A2M*, *APP*, *CLU*, *IGF2* and *PLAU* are not among the hyperlipidemia associated genes, they are retrieved only after obtaining the disease module with DIAMOND. This demonstrates that in some cases, using only the

disease associated genes is not enough to uncover the molecular mechanisms shared between diseases and using the disease module provides a more complete landscape of the disease.

Next, to retrieve the drugs directly targeting the overlapping genes (direct drugs) or their vicinity (indirect drugs), we use closeness centrality with the option of including indirect drugs (DCG-based empirical $P$ value of obtained ranked list of drugs: <0.001). We find Gemfibrozil among the top-ranked drugs (rank 6), which is originally indicated for the treatment of hyperlipidemia. Gemfibrozil is being tested in clinical trials (NCT02045056) and preclinical studies[63] give evidence of potential effectiveness of this drug for the treatment of AD. Remarkably, this drug does not directly target any of the gene products of the 7 overlapping genes, and can only be retrieved by using the indirect mode. The indirect drugs can be interpreted as drugs whose targets are closely related to the seeds; in this case, Gemfibrozil targets TTR, CYP2C8 and LPL, which interact with APOE, A2M, CLU and APP (Fig. 6), suggesting that this drug could have a positive effect by affecting several targets which altogether affect the key disease components of AD and hyperlipidemia.

## Discussion

Studies in the field of drug repurposing have so far been restricted to present either non-translational algorithms or specific predictions for certain diseases. Therefore, there is an ongoing need for integrated tools which allow experts from pharmacology or biomedical research fields to easily utilize network-based drug repurposing methods and adapt them to their individual use cases.

With NeDRex, we introduce an integrated, user-friendly platform, which allows non-computer scientists and clinicians to mine different layers of a large heterogeneous biological network —the NeDRexDB knowledgebase. NeDRex provides users with a variety of network-based methods (available via NeDRexApp) to derive disease modules associated with diseases under study and prioritize drugs directly or indirectly targeting the disease modules. NeDRex also has the feature to provide prioritization for only approved drugs, which accelerates the drug development process by skipping the pre-clinical research phase and going directly into clinical trials. Benefiting from the expert-in-the-loop paradigm, researchers from biomedical sciences can leverage their domain knowledge at different points of the workflow, e.g., by filtering disease genes already provided by the platform or by using their own sets of genes as starting points for the algorithms. NeDRex hence enables researchers and clinicians to derive disease modules, explore disease-associated mechanisms, and identify drug repurposing candidates associated with these mechanisms.

We have presented five use cases which demonstrate that NeDRex can be used to mine biologically meaningful candidate disease modules as well as potentially repurposable drugs. In particular, we have shown that by using the functionalities available in NeDRex, we can identify candidate drugs that can be further explored for the treatment of inflammatory bowel disease, pulmonary embolism, Huntington's disease, and Alzheimer's disease. All results were statistically validated by empirical $P$ values. Employing different validation methods for the use cases presented in the Results section, we computed 33 $P$ values, 29 of which were statistically significant with significance level 0.05 (lists of all computed $P$ values can be found in the Supplementary Information).

While the expert-in-the-loop paradigm is one of the main advantages of the NeDRex platform, it is also its most important limitation. When using NeDRex, investing domain knowledge is

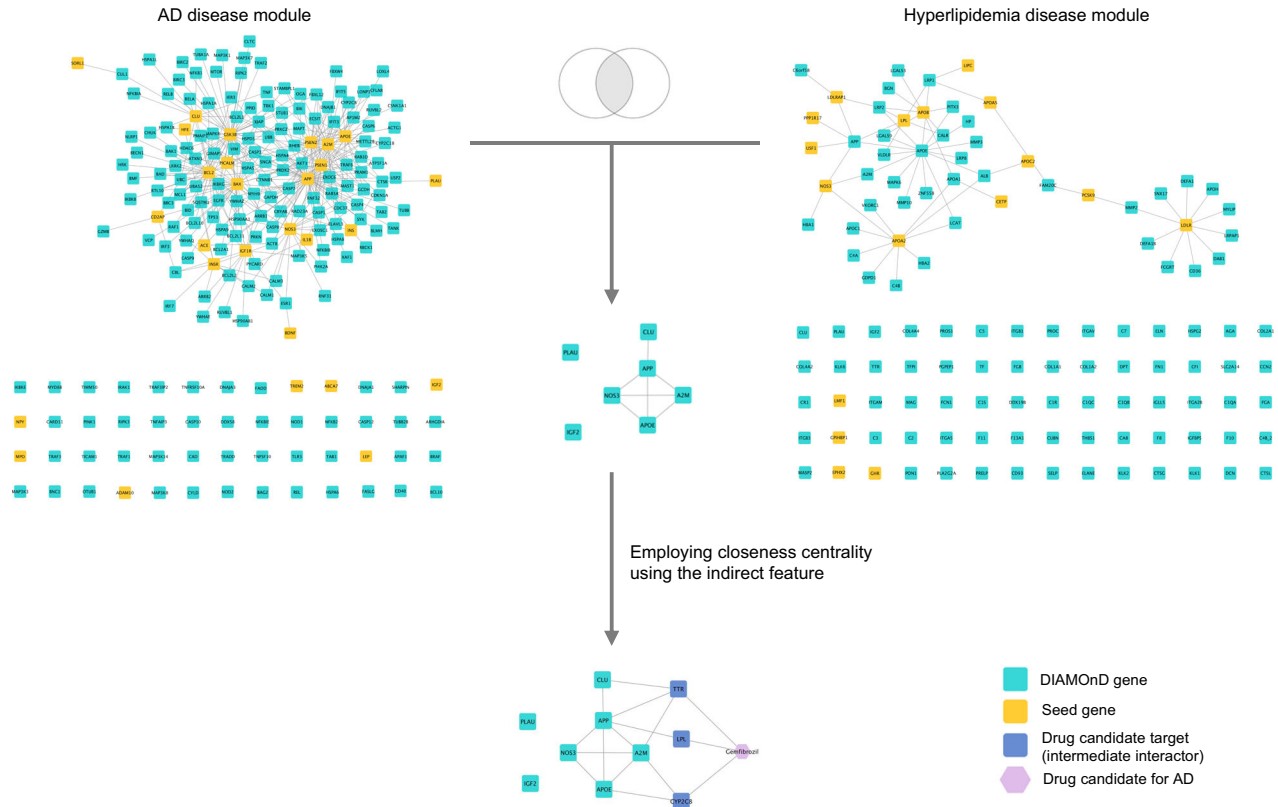

**Fig. 6 Gemfibrozil indirectly targets the intersection of disease modules.** The AD and hyperlipidemia disease modules (top left and top right, respectively) derived by DIAMOnD using the corresponding disease-associated genes (orange nodes). The intersection of the disease modules is shown in the middle. Gemfibrozil indirectly targets the intersection through TTR, CYP2C8, and LPL (bottom). To allow better visualisation, subsets of actual networks corresponding to the disease modules are shown here.

not an option but a requirement. If used blindly, obtaining biologically meaningful disease modules or promising drug repurposing candidates is unlikely. Importantly, also the empirical *P* values cannot replace the expert user, because they, too, are conditional on current knowledge (see "Methods" for details).

As stated above, NeDRex can only deliver putative drug candidates for further evaluation. Whereas the proposed drugs target proteins involved in potentially important disease mechanisms, the efficacy of the drug candidates needs to be verified by follow-up investigations and tested according to established rules and guidelines for clinical trials.

Finally, the integrated databases have their inherent limitations, which are reflected in our platform as well. Such limitations include false positive PPIs[64], literature bias due to under- and over-studied genes[65], and the fact that drug-protein associations available in the integrated databases do not distinguish between activation and inhibition.

For future versions of the database, we are planning to integrate disease symptoms and drug side effects data, which will allow investigation into different disease similarity and drug repositioning approaches. Regarding drug indications, previous studies (e.g., RepoDB[66]) include instances of failed drugs which act as false negatives for drug indications. This has a number of advantages, such as not requiring closed-world assumptions to be made, and NeDRexDB could benefit from including similar data (e.g., from ClinicalTrials.gov). Finally, we are planning to integrate further drug repurposing databases that include tissue-level gene expression information which could help to understand why specific molecular mechanisms only lead to diseases in specific tissues.

## Methods

**Data integration and construction of NeDRexDB**. NeDRexDB is a graph database that was constructed by integrating 10 source databases using a crowdsourcing framework. These 10 databases with their corresponding versions are shown in Supplementary Table 1. For all 10 databases, we wrote parsers to extract entities (nodes) and the relationships between entities (edges), and store them in a MongoDB instance. MongoDB was chosen as the database for two primary reasons; firstly, MongoDB has a flexible schema, which provides the freedom to readily add new characteristics to documents in the database, whilst simultaneously allowing selective enforcement of certain guarantees. Secondly, MongoDB provides a rich set of operations for querying and updating, which facilitates data integration. For more details about the data integration see Supplementary Information.

To facilitate integration, each entity in NeDRexDB was given a `primaryDomainId` of the form `{database}.{identifier}` (e.g., uniprot.P51587 for the Homo sapiens BRCA2 protein). In the cases of Proteins, Genes, and Pathways, all of the databases integrated here use UniProt, Entrez, and Reactome respectively, and so integration can be done simply on identifiers. For Drugs, DrugBank identifiers were chosen as the primary ID because DrugCentral tends to cross reference drugs to DrugBank identifiers.

Integration of diseases was more challenging, as there are no consistent identifiers used between different databases. Furthermore, mappings between disease identifiers in different databases are not complete, and many datasets do not have a hierarchy in disease concepts. Capturing a disease hierarchy in the NeDRexDB was important, as many diseases have very precise sub-typing which, for some analyses, may be too specific. We decided to use the Monarch Disease Ontology (MONDO) as the primary identifier for diseases, as the mapping between MONDO and other identifiers (e.g., the Unified Medical Language Systems (UMLS), used by DisGeNET) is more complete than others [https://www.disgenet.org/downloads], and includes a hierarchy.

**Accessing NeDRexDB**. The NeDRexDB can be accessed in two ways. The first is through a RESTful API, available at https://api.nedrex.net/, and the second is through a Neo4j endpoint, available at http://neo4j.nedrex.net/.

The routes from the API make a range of services available, including obtaining nodes and edges from NeDRexDB, ID mapping, and traversing the MONDO disease hierarchy. In addition, the API makes routes available for constructing

networks in graphml format based on users selected specifications. Graph construction is highly configurable, with options allowing filtering based on attributes (such as drug groups, IID evidence types, thresholds of gene-disease associations from DisGeNET). The documentation for the routes can be found at https://api.nedrex.net/. An overview of all the node and edge types available in the NeDRexDB metagraph is illustrated in Supplementary Fig. 1 and also given in Supplementary Table 2 with their corresponding numbers.

The MongoDB representation of the data was imported into a Neo4j instance, allowing users to run Cypher queries, and thus have even finer control over queries than the API allows. One major difference between the Neo4j endpoint and the API is that drugs obtained via the API are collapsed into a single Drug type by default, whereas the Neo4j instance divides these into two types, BiotechDrugs and SmallMoleculeDrugs–the abstraction used by DrugBank where drugs are sourced from.

**Network-based algorithms for disease module identification and drug repurposing**. In NeDRex, we have implemented several well-established network medicine algorithms to provide various investigation options for numerous particular medical, therapeutic, and research questions. The available algorithms are detailed below. NeDRexApp allows users to select among these algorithms. Note that, although the NeDRexDB contains also predicted PPIs, only experimentally validated PPIs are considered for the networks on which the algorithms are run.

MuST—The Steiner tree problem is an optimization problem whose objective is to find a tree of minimum cost connecting the set of seeds (terminals)[67]. For NeDRex we established a multi-Steiner trees method that aggregates several approximates of Steiner trees into a single subnetwork. By selecting genes associated with a disease under study as seeds, MuST extracts a connected subnetwork which potentially incorporates the genes involved in the disease pathways and mechanism. The motivation behind returning multiple trees instead of one is that the solutions to the Steiner tree problem are usually non-unique and computing several Steiner trees increases the stability of the extracted mechanism. Hub nodes, i.e., proteins having high number of interactions in the interactome, inherently have a higher chance of appearing in the extracted trees. In order to penalize the hubs and consequently extract mechanisms more specific to the disease of interest, users can conduct the MuST algorithm with the hub penalty parameter. This parameter incorporates the degree of neighboring nodes as edge weights in the optimization. In NeDRex, the MuST algorithm is implemented on the protein-protein layer of the heterogeneous network to obtain disease modules which could contain targets of putative drug repurposing candidates.

DIAMOnD—DIAMOnD[8] identifies a candidate disease module around a set of known disease genes (seeds) by greedily adding nodes with a high connectivity significance to the module, i.e., nodes in whose neighborhoods nodes already contained in the module are significantly overrepresented. In the iterative algorithm of DIAMOnD, the connectivity significance of all direct neighbors of seeds is computed. Then, the most significantly connected node is integrated into the module, leading to expansion of the module by one node per iteration. Subsequently, the connectivity significance is recomputed w.r.t. the updated module and the process iterates until the desired module size has been reached. In contrast to MuST, DIAMOnD does not necessarily return a connected subnetwork as the disease module. In our platform, the DIAMOnD algorithm is applied to the protein-protein layer of the integrated network to derive disease modules which could incorporate targets of potential drug repurposing candidates.

BiCoN—BiCoN is a network-constrained biclustering method that is used for integrative analysis of gene expression and PPI networks[21]. BiCoN simultaneously clusters patients and genes such that genes also form a connected subnetwork in the PPI network. As an unsupervised method, BiCoN does not need a known phenotype for patients, which allows it to find entirely data-driven patients subgroups.

Closeness centrality—Closeness centrality is a node centrality measure that prioritizes the nodes in a network based on the lengths of their shortest paths to all other nodes in the network. In NeDRex, we implemented a modified version, where closeness is calculated with respect to only the selected seeds. The motivation behind this modification is to favorably select drugs that are at a close distance to the nodes in the disease module and are hence good candidates as repurposable drugs. Our implementation focuses on the combination of protein-protein and protein-drug layers of the heterogeneous network which result in a ranked list of drugs.

TrustRank—TrustRank is a modification of Google's PageRank algorithm, where the initial trust score is iteratively propagated from seed nodes to adjacent nodes using the network topology. It prioritizes nodes in a network based on how well they are connected to a (trusted) set of seed nodes[20]. In NeDRex, it is executed on the combination of protein-protein and protein-drug layers of the heterogeneous network to obtain a ranked list of drugs that could be putative drug repurposing candidates. The damping factor parameter (range 0–1) controls the rate of trust propagation across the network. A higher damping factor returns results in a more explorative fashion.

**Statistical validation**. To validate the statistical significance of the lists of drugs and disease mechanisms returned by NeDRex, we have implemented three validation methods, each with two variations, based on empirical P values. These validation methods allow the user to assess the statistical significance of the results obtained via different algorithms available in NeDRex, and hence make the algorithms and their results assessable and comparable w.r.t. validity and relevance. As reference, all three validation methods require a list of drugs indicated for the treatment of the disease under scrutiny. This list can either be provided by the user or be obtained directly from NeDRexDB. All empirical P values depend on the quality of the list of reference drugs. If this list is incomplete or contains many false positives, the P values might be misleading. Consequently, also the P values are conditional on current knowledge and therefore cannot substitute, but merely assist, the expert in the loop. The reported P values in the Results section and Supplementary Information are rounded to three significant digits and values smaller than 0.001 were indicated correspondingly.

a) Validation of drug lists computed by NeDRex—First, a big number of, e.g., 1000 (user parameter) ranked lists of randomly selected drugs, matching the size of the drug list predicted by NeDRex, are generated. For the predicted and each of the randomly selected drug lists, we compute the discounted cumulative gain (DCG)[68] defined as $DCG = \sum_{i=1}^{n} \frac{d_i}{\log_2(i+1)}$, where $n$ is the length of the ranked list of drugs, $d_i = 1$ if the $i^{th}$ drug from the sorted list of drugs is indicated for the disease of interest and $d_i = 0$ otherwise. Subsequently, an empirical P value is computed by counting the number of random drug lists whose DCGs exceed the DCG of the drug list predicted by NeDRex. We also implemented a simplified version (overlap-based) where, instead of the DCG, the overlap $\sum_{i=1}^{n} d_i$ with the reference list is used. However, unlike the DCG-based P values, this approach ignores whether the reference drugs are found early or late in the lists of drugs. Hence, it is recommended to be used if the user wishes to ignore the drug ranks for the statistical validation.

b) Validation of disease modules computed by NeDRex—This method takes into account the role of the disease module identification step in the NeDRex drug repurposing pipeline. We generate a number of, e.g., 1000 mock modules matching the size and the number of connected components of a disease module returned by NeDRex. We set the latter constraint to keep the topology of random modules similar to the result disease module. For the disease module computed by NeDRex as well as each mock module, we define its precision as the number of reference drugs targeting the module divided by the overall number of drugs targeting the module. We then compute an empirical P value by counting the number of mock modules with higher precision values than the disease module computed by NeDRex. We have also implemented a simplified approach where we do not normalize by the overall number of targeting drugs, i.e., compare intersection sizes with the reference drugs instead of precision values as defined above. If users are more interested in inspecting the number of drugs targeting a disease module, they can use the simpler version.

c) Joint validation of disease modules and drug lists computed by NeDRex—In this approach, both steps of the drug repurposing pipeline, i.e., disease module identification and drug ranking, are taken into account for the final in silico validation of drugs. Computationally, this approach is similar to the validation method for disease modules described previously. The only difference is that we now calculate the precision for the NeDRex result as the number of reference drugs contained in the drug list computed by NeDRex divided by the overall number of drugs in the list. Analogously, we use the drug lists returned by NeDRex to calculate the intersection size for the disease module computed by NeDRex. Precision values and intersection sizes for the mock modules are determined as before.

**Implementation**. Four modules compose the NeDRex platform: (i) NeDRexDB and its constituent metagraph. Two implementations of the NeDRexDB are used: one in Neo4j and one in MongoDB. The MongoDB version of the database is populated first, as described in the data integration section, and the MongoDB version is then exported to Neo4j. Both versions of the database are used in the API implementation, leveraging the query system advantages of both platforms. (ii) The Backend including some network-based algorithms (such as DIAMOnD, BiCoN, TrustRank and closeness centrality) is implemented with Python (v. 3.7.6). DIA-MOnD was obtained from https://github.com/dinaghiassian/DIAMOnD, using the 22nd Sept 2020 commit (hash beginning 2437974). BiCoN was obtained from the Python Package Index (version 1.2.11). The ranking algorithms are implemented using the `graph-tool` library (v. 2.35). (iii) NeDRexAPI was constructed in Python 3 using the `fastapi` library (v. 0.61.0). (iv) NeDRexApp for Cytoscape 3 is written in Java (JDK 8). NeDRexApp serves as the primary frontend for the NeDRex platform. In addition, NeDRexApp can be used as a stand-alone app which provides access to some functions outside of the NeDRex ecosystem. For example, the MuST algorithm is implemented in both the backend as a Java command line tool and also in NeDRexApp (JDK 8) - the latter allows users to run MuST on any custom PPI network loaded into Cytoscape.

**Reporting summary**. Further information on research design is available in the Nature Research Reporting Summary linked to this article.

# Data availability
The authors declare that the NeDRexDB knowledgebase supporting the findings of this study are available via https://api.nedrex.net/. The construction of NeDRexDB is

described accordingly within the paper and its Supplementary Information files. The NeDRexDB knowledgebase contains information obtained from the Online Mendelian Inheritance in Man® (OMIM®) database, which has been obtained through a license from the Johns Hopkins University, which owns the copyright thereto. Use of the NeDRex dataset is governed by an End User License Agreement (available at https://nedrex.net/about.html), due to requirements of including OMIM as a source database.

The following databases are used in this study: IID (http://iid.ophid.utoronto.ca/), DrugBank (https://go.drugbank.com/), DrugCentral (https://drugcentral.org/), DisGeNET (https://www.disgenet.org/), OMIM (https://omim.org/), NCBI gene info (https://www.ncbi.nlm.nih.gov/gene), UniProt (https://www.uniprot.org/), MONDO (https://mondo.monarchinitiative.org/) and Reactome (https://reactome.org/).

## Code availability

NeDRex is a public platform built of three main components: a knowledgebase (NeDRexDB, available at http://neo4j.nedrex.net/ and https://api.nedrex.net/), a Cytoscape app (NeDRexApp, available at https://apps.cytoscape.org/apps/nedrex/), and an API (NeDRexAPI, available at https://api.nedrex.net/). The NeDRexDB, NeDRexAPI, and NeDRexApp code is openly available on GitHub repositories (https://github.com/repotrial/nedrex and https://github.com/repotrial/NeDRexApp) under the terms of the GNU General Public License, Version 3.

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

## Acknowledgements

S.S., J.S., E.A., K.F., S.C., H.H.H.W.S., J.Ba., A.W., and T.K. are grateful for financial support from REPO-TRIAL. REPO-TRIAL has received funding from the European Union's Horizon 2020 research and innovation programme under grant agreement No 777111. This publication reflects only the authors' view and the European Commission is not responsible for any use that may be made of the information it contains. J.Ba. and T.K. are grateful for financial support from BMBF grant Sys_CARE (no. 01ZX1908A) of the Federal German Ministry of Research and Education. J.Ba. was partially funded by his VILLUM Young Investigator Grant no. 13154. Contribution by J.Be. is funded by the German Federal Ministry of Education and Research (BMBF) within the framework of the e:Med research and funding concept (grant 01ZX1910D). M.S.-A. is grateful for a Ph.D. fellowship funding from CONACYT (CVU659273) and the German Academic Exchange Service, DAAD (ref. 91693321). Contribution by O.L. is funded by the Bavarian State Ministry of Science and the Arts as part of the Bavarian Research Institute for Digital Transformation. A.I.C. is currently financially supported by the DFG Walter Benjamin Program (ref. DFG CA 2642/1-1). Figures 1 and 2 are created with BioRender.com.

## Author contributions

S.S., J.S., D.B.B., J.Ba., A.W., and T.K. conceived the idea and designed the platform. S.S., J.S., J.Be., E.A., G.G., K.F., S.C., T.K. performed the acquisition, harmonization and integration of databases. S.S. and D.B.B. developed and adapted the network-based algorithms for drug repurposing. S.S., E.A., G.G., M.S.-A., O.L., C.N., and A.I.C. discovered and approved the use cases. J.S. implemented the API. S.S. and J.Be. implemented the Cytoscape app. All authors provided critical feedback and discussion, assisted in the interpretation of data and use cases, writing the manuscript, and the improvement of the platform.

## Funding

## Competing interests

The authors declare no competing interests.
