## [Peer Review File · Nature Communications]

Reviewers' Comments:

Reviewer #1:

Remarks to the Author:

NeDRexDB integrates network data with gene annotation information and implements a set of algorithms to identify modules within the networks. The main purpose of the tool is drug repurposing. A session starts with selecting a set of seed genes / proteins. Those can have certain predefined properties (eg being associated with a specific disease) or defined by the user. In general, given its flexibility I believe that this tool is a useful contribution to the community.

1. The authors give several use cases. Those constitute a large fraction of the Results section. My main concern is that the authors neither aim to estimate the impact of the network algorithm, nor control for the impact of the known problems with biological networks (mainly biases and high error rates - as the authors acknowledge in the Discussion).

Just as an example (but this applies basically to all use cases) Figure 3B: how surprising is it that the resulting network is enriched in functions related to ovarian biology and cancer (eg compared to considering all interactors of the chosen nodes)? How crucial is the choice of the algorithm itself? Are the enrichments even significant? What about the functional enrichment of the connecting nodes only?

2. Sometimes statistical tests are missing to quantify the degree of surprise associated with observations made. Eg: "among the 50 top-ranked drugs 250 we find three drugs that are prescribed to alleviate the symptoms of HD" (line 250/251). What's the probability of finding these 3 drugs by chance?

3. A few formulations in the paper are overstating. Eg " we have shown that by using the functionalities available in NeDRex, we can identify potential drugs for the treatment of (...)". Given the points before (missing statistics and lack of control for biases), this needs to be toned down.

4. What's the intended update cycle of the knowledge base?

Reviewer #2:

Remarks to the Author:

Sadegh and co-workers present an integrated platform for network-based disease module discovery and drug repurposing, named NeDRex. NeDRex integrates ten different data sources covering genes, drugs, drug targets, disease annotations, and allows for network construction and analysis. Overall, this work is well-written and will contribute to the field of pharmacology or biomedical research. However, from my perspective, I don't think this manuscript is suitable for publication in Nature Communications due to lack of enough novelty. The platform seems a 'toolset' of existing databases and algorithms rather than an innovative platform. The integrated data is extracting from public available database (e.g. DrugBank, NCBI..), lacks of your own data obtained from other sources such as literature mining or omics analysis. The network algorithms implemented into the platform also ready-made approaches without original network method, which reduces the innovation of the work. In the "Results" section, the authors demonstrated several cases of the application of NeDRex, including drug discovery, drug repurposing, target identification, and pathway enrichment. However, the authors seem only quoted some literature to prove these in silico predictions, which may a bit of unconvincing. A wet-lab validation or support from large-scale clinical evidence is absolutely needed, which could better illustrate the reliability and effectiveness of the platform.

Reviewer #3:

Remarks to the Author:

This is a well-written manuscript that describes a useful new bioinformatics tool, NeDRex, that can be used to find and explore biological networks related to human disease and to identify potential repurposable drugs for treatment of a disease. The system consists of a series of graph databases that have been derived from large public datasets, a series of conveniently-packaged third-party graph analysis algorithms, a scriptable API, and user interfaces accessible either via the web or the popular biological network analysis environment Cytoscape. The authors describe the tool in an

easily accessible manner, and provide walkthroughs of several use cases illustrating the system's flexibility and utility. I checked NeDRex's online resources briefly and confirmed that the databases are up and running and that the Cytoscape app would install and run. These steps worked as advertised. I did not attempt to walk through the use case steps due to time constraints, but I did confirm that the online tutorial was accessible and that I could follow the first few steps. This is a tool that I would definitely like to apply to my own research.

The challenge of a paper like this one is convincing the reader that the tool is fit for purpose; in this case, that the top-ranked drugs it identifies do indeed alter the targeted disease pathways. Ideally one would like to see a true biological validation, for example showing that an unexpectedly large proportion of the top-ranked drugs alters the course of a mouse model of the disease. This can be difficult and time-consuming in the absence of a collaboration with a lab that has been working with the model for some time. In the absence of this type of evidence, the authors have relied on hand literature searches to demonstrate that NeDRex is able to rediscover relevant drugs and disease modules. This data is presented anecdotally, and while I was pretty well convinced that these rediscoveries were unlikely to have been found by chance, it would have been more convincing for the authors to have performed the experiment in an unbiased and quantitative fashion. For example, I can envision an experiment in which the authors first hand-curate the biomedical literature to identify all drugs that have been applied to huntington's disease (HD), freeze this list, and then measure the frequency with which these drugs appear among the top-ranked drugs identified by NeDRex as compared to similarly-sized lists of drugs selected from the database randomly.

Another question readers will have is the maintenance of the resource. NeDRex relies on the extraction of data from 9 online databases, each of which is actively maintained and constantly evolving. It is well documented that the use of outdated annotations can have a disproportionate impact on the inference of biological pathways (Wadi et al, DOI 10.1038/nmeth.3963), and the manuscript should describe the commitment of the authors to updating the underlying NeDRex databases and the frequencies with which they intend to do so. Long-term maintenance of the software is also an issue, and for this reason I strongly recommend that the authors release the full NeDRex codebase under an open source license that allows for redistribution with attribution, rather than the described terms of "available upon reasonable request."

A minor technical error that I encountered is that the among the examples used to validate the disease pathways discovered in ovarian cancer (Use Case #1) are estrogen and progesterone pathways involved in "oocyte maturation." Assuming that the authors were working on a gene set enriched in serous ovarian cancer, they should be aware that this disease does not derive from the oocyte germ cell lineage, but from the epithelial cells that line the fallopian tube and/or ovary surface. These cells are also sex hormone responsive, so it does not change the validation result, but researchers familiar with ovarian cancer will spot the mistake. If the authors have not already done so, I would recommend that they review the results of each use case with domain experts in order to spot any similar goofs.

Lastly, there is a question of novelty. This manuscript does not describe any new fundamental algorithms, analytic workflows or discoveries, but repurposes and repackages existing data and methods in a convenient way that will be powerful in the hands of expert users. In its current state this manuscript might be more suitable as an Application Note in a journal such as Bioinformatics.

We would like to thank the reviewers for their very constructive and helpful comments on our manuscript entitled “Network medicine for disease module identification and drug repurposing - The NeDRex platform”.

We addressed all your comments and believe that our manuscript was substantially improved. Below, we detail how our revision addresses your feedback. In the revised version of the manuscript and the Supplementary Information document, all changes are marked in blue.

Reviewer 1:

Summary: NeDRexDB integrates network data with gene annotation information and implements a set of algorithms to identify modules within the networks. The main purpose of the tool is drug repurposing. A session starts with selecting a set of seed genes / proteins. Those can have certain predefined properties (eg being associated with a specific disease) or defined by the user. In general, given its flexibility I believe that this tool is a useful contribution to the community.

We are happy that reviewer 1 believes that our tool is a useful contribution to the community and that the flexibility of our tool is acknowledged. We sincerely appreciate you for providing the positive and constructive comments and suggestions on our manuscript.

R1.C1) The authors give several use cases. Those constitute a large fraction of the Results section. My main concern is that the authors neither aim to estimate the impact of the network algorithm, nor control for the impact of the known problems with biological networks (mainly biases and high error rates - as the authors acknowledge in the Discussion).

Just as an example (but this applies basically to all use cases) Figure 3B: how surprising is it that the resulting network is enriched in functions related to ovarian biology and cancer (eg compared to considering all interactors of the chosen nodes)? How crucial is the choice of the algorithm itself? Are the enrichments even significant? What about the functional enrichment of the connecting nodes only?

Our reply

We fully agree with the reviewer. In particular the choice of algorithm can affect the results. This is a general problem of the systems medicine field, however, and a systematic evaluation of different algorithms is beyond the scope of this paper. We instead refer the readers to recent papers describing comprehensive benchmarks of network-based algorithms for disease module identification

doi.org/10.1038/s41592-019-0509-5,
doi.org/10.1371/journal.pcbi.1007276).

doi: [10.1038/srep46598](https://doi.org/10.1038/srep46598),

Importantly, no single algorithm is the best performer in all cases, in addition to the impact of the algorithms' hyperparameters. Hence, NeDRex employs a human-in-the-loop approach as the most suitable algorithm depends on the diseases under study and the starting/input seeds. Most importantly, though, we are very grateful for the suggestions of the reviewers regarding the statistical validation of the results, which are meant to allow the users to judge quantitatively the performance of the algorithms for their individual use cases. We address how this was implemented in the response to the reviewer's second comment (R1.C2).

Regarding Figure 3B, the functional enrichment analysis itself already takes into account the background frequency (the total number of genes annotated in a term); thus, the results related to ovarian biology and cancer are significant (depicted by the circle size). We found significant results with seeds only and seeds with connectors; however, there were no significant functions when the analysis was performed with the connectors only. In fact, the increase in the number of pathways associated with the entire module (seeds+connectors) compared to the functions found for seed genes only demonstrates the value of the network-based algorithms to extract active modules.

The reviewer also mentioned an approach considering the interactors of the seeds. However, this method could return a very high number of nodes. For instance, we found 1,741 interactors (neighbors) of the eight OC seed genes in the NeDRexDB interactome. Including such a large list of genes in the pathway enrichment analysis results in a high number of pathways and can confound biological interpretation.

R1.C2) Sometimes statistical tests are missing to quantify the degree of surprise associated with observations made. Eg: "among the 50 top-ranked drugs 250 we find three drugs that are prescribed to alleviate the symptoms of HD" (line 250/251). What's the probability of finding these 3 drugs by chance?

Our reply

Excellent suggestion! Implementing statistical validation methods has significantly improved NeDRex as a tool and the impact of the paper. We have now introduced and implemented three methods, each with two variants, to systematically validate the results obtained by our pipeline statistically. All approaches have been implemented as new features to a new version of our NeDRexApp and will provide the users with greater confidence in their results. Details are described below. We also utilized these

newly implemented NeDRex features to evaluate each of our use cases (i.e. we provide *P*-values).

- a) As the reviewers suggested, we have implemented a statistical approach for the validation of drugs returned as result by NeDRex, where we first collect the list of drugs indicated for the treatment of the disease and use this list as the true reference list of drugs. This list can be assembled automatically from the NeDRex database, but - in case there are only a few known drugs for the treatment of the disease - the reference list can be extended by drugs in clinical trials, for instance, or extended manually. Then, we derive a user-given number, e.g. 1000, lists of randomly selected drugs that match the size of the drug list predicted by NeDRex. We estimate the significance by calculating an empirical *P*-value by counting the number of random lists having larger overlap with the reference list of drugs than that of the NeDRex result list. Furthermore, we implemented a variation of this method where the ranks of the reference drugs in the output are also considered. We define discounted cumulative gain (DCG) for

a list of ranked drugs as $DCG = \sum_{i=1}^n \frac{d_i}{\log_2(i+1)}$, where n is the length of the ranked

list of drugs, $d_i = 1$ if the i^{th} drug from the sorted list of drugs is indicated for the disease of interest and $d_i = 0$ otherwise. The DCG metric captures whether the true list of drugs (reference drugs) were retrieved early or late in the ranked list. The rank-based empirical *P*-value is computed by counting the number of random drug lists with DCG values higher than that of the NeDRex result list.

- b) We implemented an approach for the statistical validation of the reported disease modules. It takes the role of the disease module identification step (MuST, DIAMOnD, BiCoN algorithms) in our drug repurposing pipeline into account. We generate a user-given number (for our paper 1000) mock modules (mechanisms) that match the size and the number of connected components of a disease module returned by NeDRex (for details refer to the updated “Methods” section of the manuscript). For each mock module, we obtain the list of drugs targeting the nodes in the subnetwork. We then compute an empirical *P*-value by counting the number of mock modules whose drug lists have larger overlap with the reference list of drugs than that of the NeDRex output. We also provide users a more conservative version of this approach, where we normalize the number of overlapping drugs by the total number of drugs for each mock module, i.e. comparing the precision rather than the size of the overlaps.
- c) Furthermore, we implemented a method for the joint validation of disease modules and drug lists computed by NeDRex. In this approach, both steps of the

drug repurposing pipeline, i.e. disease module identification and drug ranking, are taken into account. Computationally, this approach is similar to the validation method for disease modules described above. The difference is that we now compute the precision for the NeDRex result as the number of reference drugs contained in the drug list computed by NeDRex divided by the overall number of drugs in the computed list. Analogously, we use the drug lists returned by NeDRex to compute the intersection size for the disease module computed by NeDRex. Precision values and intersection sizes for the mock modules are computed as before.

We report all empirical P -values, computed with different approaches, for all use cases now in the Supplementary Information. However, in the main paper we only give the P -values computed, in our opinion, with the most balanced approach. In total, we computed 33 P -values, out of which 29 were statistically significant to a significance level of 0.05.

Specifically for the HD use case that was asked by the reviewer, the DCG-based empirical P -value of the ranked list of drugs computed using TrustRank is 0.011 and the precision-based P -value obtained by joint validation of module and drug list is 0.048.

R1.C3) A few formulations in the paper are overstating. Eg " we have shown that by using the functionalities available in NeDRex, we can identify potential drugs for the treatment of (...)". Given the points before (missing statistics and lack of control for biases), this needs to be toned down.

Our reply

The statistical significance of the NeDRex output based on the use cases is now added in the main manuscript as well as the Supplementary Information document. Nevertheless, we have slightly modified our statement as suggested by the reviewer: "we can identify candidate drugs that can be further explored for the treatment of ..."

R1.C4) What's the intended update cycle of the knowledge base?

Our reply

The update cycle is automatic and monthly for all databases and weekly for OMIM - in accordance with the OMIM license policy (more information can be found in the Supplementary Information).

Reviewer 2:

Summary: Sadegh and co-workers present an integrated platform for network-based disease module discovery and drug repurposing, named NeDRex. NeDRex integrates ten different data sources covering genes, drugs, drug targets, disease annotations, and allows for network construction and analysis. Overall, this work is well-written and will contribute to the field of pharmacology or biomedical research.

We are grateful to the reviewer for acknowledging the contribution of our tool to the research field. We have incorporated the changes to reflect the suggestions provided by the reviewer.

R2.C1) However, from my perspective, I don't think this manuscript is suitable for publication in Nature Communications due to lack of enough novelty. The platform seems a 'toolset' of existing databases and algorithms rather than an innovative platform. The integrated data is extracting from public available database (e.g. DrugBank, NCBI..), lacks of your own data obtained from other sources such as literature mining or omics analysis. The network algorithms implemented into the platform also ready-made approaches without original network method, which reduces the innovation of the work.

Our reply

Novelty 1: Integration of the NeDRexDB knowledgebase with network medicine algorithms

Part of our ethos while developing the NeDRex platform has been “the whole is greater than the sum of its parts”. One novelty of NeDRex lies in our rich, unified knowledgebase combined with the user-friendly and intuitive algorithm implementations and pipelines, which facilitate interpretation by integrating ways of exploring and visualising disease clusters.

NeDRexDB integrates crucial information that is disparate across public databases to allow the user to derive links that were previously unattainable other than through scrupulous literature and data mining exercises. The knowledgebase consists of a novel compilation of several datasets integrated at a high mapping standard, including at the disease ontology level. Crucially, there is no shrinkage of the disease lists to incorporate only the ones that can be manually mapped, but we incorporate as many high confidence, automatic disease mappings as possible.

NeDRex additionally packages complex algorithms (such as an enhanced version of MuST developed by our team (doi.org/10.1038/s41467-020-17189-2)), and guides the user through simple, novel task pipelines to extract information and results pertinent to

their field of interest. Finally, the results can (thanks to the reviewers' suggestions) be statistically validated via the newly implemented validation approaches explained above. Unlike all existing methods and tools, NeDRex hence covers the entire workflow of network-guided drug repurposing and hence renders the network medicine paradigm easily accessible to researchers from the biomedical sciences without strong expertise in computer science.

Novelty 2: Incorporation of the expert-in-the-loop paradigm

As acknowledged by the reviewers, the system also benefits heavily from an effective expert-in-the-loop scenario. Clinicians and researchers who have a detailed knowledge on the disorder or pathway that they are researching can quickly categorise and prioritise information that is highlighted by the interactive NeDRexApp. Thus, this unique system can output context-specific results more biologically meaningful for downstream analyses. The majority of systems medicine algorithms are usually provided separately (e.g. as individual Python, Java, or R packages) and are not readily available in the expert-in-the-loop and analysis workflow form, which constitutes a significant bottleneck for clinicians, pharmacologists, and biologists.

To our knowledge, the only drug repurposing tools comparable to NeDRex are Hetionet ([doi:10.7554/eLife.26726](https://doi.org/10.7554/eLife.26726)), PROMISCUOUS 2.0 (doi.org/10.1093/nar/gkaa1061) and CoVex (doi.org/10.1038/s41467-020-17189-2). However, each resource offers much less flexibility and features than the NeDRex platform that we have designed. In particular, CoVex is limited to COVID-19 drug repurposing. Hetionet consists of a fixed set of 137 diseases and algorithms are provided as separate Python packages rather than integrated into the platform. Hetionet is static and allows the user only to browse precomputed results. Similarly, a fixed set of candidate drugs is precomputed within PROMISCUOUS 2.0.

We believe that the integration, flexibility to incorporate expert knowledge, customizability, and full analysis pipeline and toolset itself provided by the NeDRex platform represents the way forward for systems medicine resources and tools, bringing *in silico* prediction methods closer to clinical translation.

R2.C2) In the "Results" section, the authors demonstrated several cases of the application of NeDRex, including drug discovery, drug repurposing, target identification, and pathway enrichment. However, the authors seem only quoted some literature to prove these *in silico* predictions, which may a bit of unconvincing. A wet-lab validation or support from large-scale clinical evidence is absolutely needed, which could better illustrate the reliability and effectiveness of the platform.

Our reply

As noted by the reviewers and the editor, experimental validation of our top-ranked list of drugs will be time consuming and hard to achieve. Therefore, we have devised three main methods to systematically validate the results obtained by our pipeline statistically. All approaches for statistical validation have been implemented as new features in our NeDRexApp and will provide the users with greater confidence in their results. Details on these approaches together with a summary of computed empirical *P*-values for all use cases are described in our reply to comment **R1.C2**.

Reviewer 3:

Summary: This is a well-written manuscript that describes a useful new bioinformatics tool, NeDRex, that can be used to find and explore biological networks related to human disease and to identify potential repurposable drugs for treatment of a disease. The system consists of a series of graph databases that have been derived from large public datasets, a series of conveniently-packaged third-party graph analysis algorithms, a scriptable API, and user interfaces accessible either via the web or the popular biological network analysis environment Cytoscape. The authors describe the tool in an easily accessible manner, and provide walkthroughs of several use cases illustrating the system's flexibility and utility. I checked NeDRex's online resources briefly and confirmed that the databases are up and running and that the Cytoscape app would install and run. These steps worked as advertised. I did not attempt to walk through the use case steps due to time constraints, but I did confirm that the online tutorial was accessible and that I could follow the first few steps. This is a tool that I would definitely like to apply to my own research.

We appreciate the time and effort that the reviewer has dedicated to providing this positive feedback as well as insightful and constructive suggestions on our manuscript.

R3.C1) The challenge of a paper like this one is convincing the reader that the tool is fit for purpose; in this case, that the top-ranked drugs it identifies do indeed alter the targeted disease pathways. Ideally one would like to see a true biological validation, for example showing that an unexpectedly large proportion of the top-ranked drugs alters the course of a mouse model of the disease. This can be difficult and time-consuming in the absence of a collaboration with a lab that has been working with the model for some time. In the absence of this type of evidence, the authors have relied on hand literature searches to demonstrate that NeDRex is able to rediscover relevant drugs and disease modules. This data is presented anecdotally, and while I was pretty well convinced that these rediscoveries were unlikely to have been found by chance, it would have been

more convincing for the authors to have performed the experiment in an unbiased and quantitative fashion. For example, I can envision an experiment in which the authors first hand-curate the biomedical literature to identify all drugs that have been applied to huntington's disease (HD), freeze this list, and then measure the frequency with which these drugs appear among the top-ranked drugs identified by NeDRex as compared to similarly-sized lists of drugs selected from the database randomly.

Our reply

Excellent suggestion! Implementing statistical validation methods has significantly improved NeDRex as a tool and the impact of the paper. We have now introduced and implemented three main methods, to systematically validate the results obtained by our pipeline statistically. Specifically, method “a” is devised based on your proposed approach. All approaches have been implemented as new features to a new version of our NeDRexApp and will provide the users with greater confidence in their results. Details are described in our reply to comment **R1.C2**. We also utilized these newly implemented NeDRex features to evaluate each of our use cases (i.e. we provide *P*-values).

R3.C2) Another question readers will have is the maintenance of the resource. NeDRex relies on the extraction of data from 9 online databases, each of which is actively maintained and constantly evolving. It is well documented that the use of outdated annotations can have a disproportionate impact on the inference of biological pathways (Wadi et al, DOI 10.1038/nmeth.3963), and the manuscript should describe the commitment of the authors to updating the underlying NeDRex databases and the frequencies with which they intend to do so. Long-term maintenance of the software is also an issue, and for this reason I strongly recommend that the authors release the full NeDRex codebase under an open source license that allows for redistribution with attribution, rather than the described terms of "available upon reasonable request."

Our reply

The update cycle is automatic and monthly for all databases and weekly for OMIM - in accordance with the OMIM license policy (more information can be found in the Supplementary Information).

With regard to long term maintenance of the software, NeDRex is currently supported by the H2020 project REPO-TRIAL and is the main bioinformatics result. NeDRex is co-hosted by five universities (University of Hamburg, Newcastle University, University of Erlangen, Technical University of Munich as well as Technische Universität Braunschweig) who are all dedicated to maintenance of the platform and have dedicated in-house funding to support maintenance as well as future developments in

the long run. Also, the International Network Medicine Alliance (where senior author Baumbach is steering board member and co-founder) will support maintaining NeDRex. We have also released the source code under the GPL3 license, and the following is the Github repository link: <https://github.com/repotrial/nedrex>.

R3.C3) A minor technical error that I encountered is that among the examples used to validate the disease pathways discovered in ovarian cancer (Use Case #1) are estrogen and progesterone pathways involved in "oocyte maturation." Assuming that the authors were working on a gene set enriched in serous ovarian cancer, they should be aware that this disease does not derive from the oocyte germ cell lineage, but from the epithelial cells that line the fallopian tube and/or ovary surface. These cells are also sex hormone responsive, so it does not change the validation result, but researchers familiar with ovarian cancer will spot the mistake. If the authors have not already done so, I would recommend that they review the results of each use case with domain experts in order to spot any similar goofs.

Our reply

The use case #1 is not focused on serous ovarian cancer (SOC) only, but on ovarian cancer (OC) in general. We reviewed the results of each use case as suggested.

Regarding the specific case mentioned by the reviewer: Despite SOC's high frequency among OC patients, the seed genes (from OMIM, DisGeNET and MONDO) are not exclusive to SOC. Instead, the genes provided can be associated with other malignancies of the ovary. This is the reason why the pathways and gene ontology terms are not uniquely associated with epithelial processes, but also to other functions of the ovary and also why we maintained the use of "ovarian cancer" and not "serous ovarian cancer" nomenclature in this use case.

R3.C4) Lastly, there is a question of novelty. This manuscript does not describe any new fundamental algorithms, analytic workflows or discoveries, but repurposes and repackages existing data and methods in a convenient way that will be powerful in the hands of expert users. In its current state this manuscript might be more suitable as an Application Note in a journal such as Bioinformatics.

Our reply

The novelty aspects of our platform is described in detail in our reply to comment **R2.C1**.

Reviewers' Comments:

Reviewer #1:

Remarks to the Author:

I appreciate the effort the authors did to address my points. The manuscript and the tool have been improved. However, I think my main concern is still valid: the tool described in the manuscript is certainly a useful contribution to the community. However, its flexibility to use different algorithms and networks makes it very difficult to assess the relevance and validity of the use cases without more stringent controls and comparisons.

I think it's a fine manuscript as it stands. However, the room that the use cases take and the way they are presented are somewhat misleading.

Reviewer #3:

Remarks to the Author:

The authors have greatly improved both the utility of the NeDRex tool and addressed my major criticisms by implementing three complementary statistical tests to measure the significance of associations uncovered by the tool. This is a major improvement to the tool. My other major concern, the lack of algorithmic novelty, was addressed effectively in the authors' rebuttal.

In light of the responsiveness of the authors to my and the other reviewers' comments, I'm now in favor of a decision to accept the paper to Nature Comm.

We would like to thank the reviewers for their comments on our manuscript entitled “Network medicine for disease module identification and drug repurposing - The NeDRex platform”.

We addressed the remaining concerns of the reviewers as outlined in the following. In the revised version of the manuscript, all changes are marked in blue.

Reviewer 1:

I appreciate the effort the authors did to address my points. The manuscript and the tool have been improved. However, I think my main concern is still valid: the tool described in the manuscript is certainly a useful contribution to the community. However, its flexibility to use different algorithms and networks makes it very difficult to assess the relevance and validity of the use cases without more stringent controls and comparisons.

I think it's a fine manuscript as it stands. However, the room that the use cases take and the way they are presented are somewhat misleading.

Our reply

To address this concern, we have added new methods to judge the statistical significance of the results provided by the different algorithms. We believe that these make the algorithms and their results w.r.t. relevance and validity assessable. This is now explicitly stated in the manuscript's methods section. We now also stress in the results section that the use cases are only meant to exemplarily illustrate the practical relevance and utility of NeDRex in different and concrete scenarios, and that the results reported for the use cases merely constitute hypotheses which have not been validated experimentally. To further emphasize the fact that the NeDRex platform itself, rather than the individual use cases, constitutes the main result presented in our article, we added a table to the results section summarizing the platform's most important features (Table 1 in the revised manuscript).